# Influence of TIG and Laser Welding Processes of Fe-10Cr-4Al-RE Alloy Cracks Overlayed on 316L Steel Plate

**DOI:** 10.3390/ma15103541

**Published:** 2022-05-15

**Authors:** Bin Xi, Bo Liu, Song Li, Disi Wang, Youpeng Zhang, Peter Szakálos, Jesper Ejenstam, Janne Wallenius, Guangqing He, Wenyang Zhang

**Affiliations:** 1Institute of Modern Physics, Fudan University, Shanghai 200433, China; 19110200027@fudan.edu.cn (B.X.); 20210200011@fudan.edu.cn (B.L.); 21210200010@fudan.edu.cn (S.L.); 20210200015@fudan.edu.cn (D.W.); heguangqing2013@163.com (G.H.); 2Division of Surface Chemistry and Corrosion Science, Royal Institute of Technology, 10691 Stockholm, Sweden; peter@leadcold.com; 3Sandvik Heating Technology AB, Kanthal, 73427 Hallstahammar, Sweden; jesper.ejenstam@sandvik.com; 4Division of Nuclear Engineering, Royal Institute of Technology, 10691 Stockholm, Sweden; janwal@kth.se; 5Shanghai Electric Group, Shanghai 200233, China; zhangwy2@shanghai-electric.com

**Keywords:** CiADSvessel, FeCrAl weld overlay, austenitic stainless steel plate, mechanical tests

## Abstract

In this paper, the possibility of applying different welding strategies to overlay an FeCrAl layer against corrosion from heavy liquid metal on a plain plate made of 316L austenitic stainless steel was investigated. This technology could be used in manufacturing the main vessel of CiADS, which may be considered as a more economic and feasible solution than production with the corrosion-resistant FeCrAl alloy directly. The main operational parameters of the laser welding process, including laser power, weld wire feeding speed, diameter of the welding wire, etc., were adjusted correspondingly to the optimized mechanical properties of the welded plate. After performing the standard nuclear-grade bending tests, it can be preliminarily confirmed that the low-power pulse laser with specific operational parameters and an enhanced cooling strategy will be suitable to surface an Fe-10Cr-4Al-RE layer with a thickness of approximately 1 mm on a 40 mm-thick 316L stainless steel plate, thanks to the upgraded mechanical properties incurred by refined grains with a maximum size of around 300 μm in the welded layer.

## 1. Introduction

Pb-based alloys are eutectics or alloys formed by adding lead as the basic material and other elements as supplementaries. The melting point of Pb-based alloys is lower than that of lead, and other physical properties are similar to those of pure Pb. Lead–bismuth eutectic (LBE) is the most commonly used alloy in the field of nuclear power, in the form of fission reactors [1]. The mass ratio between Pb and Bi in LBE is 44.5:55.5, which provides the lowest melting point among all lead–bismuth alloys [2].

The main advantages of Pb-based materials as coolants are the following: (1) due to the relatively high melting point of Pb-based materials, the Pb-based reactor could operate at normal pressure, leading to a much thinner vessel wall [3] and lower possibility of the loss of coolant accident (LOCA); (2) due to the high boiling point and high thermal conductivity of Pb-based coolant materials, Pb-based reactors can work at relatively high temperatures, which may greatly improve the system’s energy efficiency; (3) due to the relatively small cross-sections for the neutron capture and scattering of Pb-based materials, a harder neutron energy spectrum could be produced in Pb-based reactors, which may improve the neutron economy and, therefore, leave extra neutrons for potential transmutation or breeding; (4) thanks to the high density of LBE, significant natural convection could be expected after losing the driving force from primary pumps [4]; (5) due to the good chemical inertness of LBE, there will be no violent chemical reactions during contact with water and air; (6) Pb-based materials can form compounds with volatile radioactive iodine and cesium nuclides, which may reduce the radioactive inventory in reactors.

In addition to the common characteristics stated above, different types of Pb-based materials, such as lead, lead–bismuth, and lead–lithium, have their own characteristics and were proposed for different types of reactors. LBE was chosen as the primary coolant material of the China-initiated Accelerator-Driven System (CiADS) [5], thanks to its low melting temperature, high boiling temperature, chemical inertness and neutron transparency. However, its corrosive property due to the dissolution of elements was reported to cause significant damage to the micro-structure of the alloy, especially at a high temperature and under severe neutron irradiation [6,7,8]. The corrosion effect from heavy liquid metal on stainless steel was reported to be highly affected by the oxygen content in the heavy liquid metal. Proper setting of the oxygen content in liquid is, therefore, able to permit the rapid formation of a protective oxide layer on the alloy surface, while also permitting less oxide impurity production. Generally, the introduction of aluminum into ferritic alloys may lead to better resistance to potential heavy liquid metal corrosion, thanks to the formation of protective alumina [9,10,11,12,13,14]. Due to their excellent properties, such as low cost, heat resistance and corrosion resistance, FeCrAl alloys have a wide range of applications under conditions of high temperature and they require a high level of corrosion resistance [15,16].

As the first barrier in a nuclear fuel, the choice of an accident tolerant fuel (ATF) is a very important issue in the field of nuclear fuel development [17]. After the Fukushima accident, there has been an urgent need to develop suitable fuel cladding materials to meet the performance requirements under severe accident conditions, which will, thereby, greatly improve the accident tolerance capabilities of future nuclear power plants [18,19,20]. Advanced cladding materials that meet these requirements can provide stronger oxidation resistance and better physical properties during severe accidents, such as SBO, LOCA, etc. [21]. This advanced cladding material must first be able to achieve the same performance as the general cladding material under the normal working condition.

In recent years, in order to solve the problem of selecting ATF cladding materials, many related studies were carried out on some candidate cladding materials. Qiu et al. [12] has made some research progress in several cladding candidate materials, such as SiC, FeCrAl and the MAX phase material Ti_3_SiC_2_. They also evaluated the physical properties of these cladding candidates in terms of thermal hydraulics and mechanical mechanics, aiming to better understand and simulate the behavior of these cladding materials.

FeCrAl alloy has the characteristics of high temperature resistance and corrosion resistance, so it has become one of the best candidate materials for ATF. Wu et al. [13] used the Monte Carlo calculation tool Serpent and fuel performance program BISON from INL to perform neutronics and fuel performance analyses of ATF. The research results show that the application of FeCrAl cladding material can significantly improve the cladding performance, such as flattening the axial temperature distribution and delaying the gap closure [17]. Additionally, hydrogen permeability was obtained for several FeCrAl alloys using a static permeation platform. The results were compared, calibrated and verified with traditional 304 stainless steel. In addition, they discussed the possibility of a protective alumina layer on the surface of FeCrAl cladding as a tritium barrier. Han et al. [15] introduced Mo as a buffer layer in the cladding, which significantly reduced the interdiffusion of Zr and Fe [22]. It was also demonstrated that a 7 μm-thick FeCrAl coating could provide excellent resistance to oxidation at temperatures up to 1200 °C [23,24]. The working limit was, therefore, significantly higher than that of traditional 316L austenitic stainless steel [25,26,27].

Based on the research results in the above publications, we can preliminarily conclude that FeCrAl alloy materials with excellent high temperature resistance and corrosion resistance could be considered as candidate materials for Pb-based reactors.

Ferritic FeCrAl alloys are normally used as heating elements, as they areable to work at 900–1400 °C [28]. The high Cr content in these alloys was reported to cause embrittlement at around 475 °C, due to phase transformations [29,30,31]. Therefore, the embrittlement characteristic of FeCrAl alloy could be minimized by reducing the Cr content. However, the introduction of Cr was reported to be able to enhance the formation of a protective Al-rich oxide layer [30]. Among FeCrAl alloys with different elementary compositions or manufacturing processes, the Fe-10Cr-4Al-RE developed by Ejenstam et al. [30,31] was reported to be able to withstand heavy liquid metal corrosion at temperatures higher than 900 °C. However, the weldability and machinability of Fe-10Cr-4Al-RE on parts made of austenitic stainless steel still needs to be verified, due to the high content of Cr and Al in it.

Tungsten inert gas (TIG) welding is an inert gas-protected fusion welding process. The heat required during the welding process is provided by an intense electric arc, which will be generated between a tungsten electrode and work pieces [9]. Compared to the traditional welding techniques, the major merits of Ar-protected TIG welding are as follows:(1)Ar gas can prevent harmful effects from some elements in air (such as oxygen, nitrogen, hydrogen, etc.), on the arc and in the molten pool. This may reduce the burning of alloy elements and, therefore, lead to a dense, spatter-free, high-purity welding joint;(2)The arc in the Ar-protected TIG welding is stable, which may lead to the concentration of heat and, hence, ahigher arc column temperature. A higher welding efficiency, smaller heat-affected zone, lower thermal stress, less heat deformation and smaller cracks could, therefore, be expected;(3)Argon-protected TIG welding can be applied on almost all metals, especially those that are hard to melt or easily oxidized.

Laser welding technology is a surface coating technology involving different research fields, such as physics, chemistry, optoelectronics, materials, and so on. Compared with other surface treatment technologies, including TIG welding, laser coating technology has many advantages [10], such as the following:(1)The thermal deformation during welding is small, which is mainly because of the smaller heat input. The heat-affected zone on the substrate will be relatively small;(2)The cooling after the welding process is fast, which may lead to finer grains in the melt zone.The micro-structure will then be relatively dense, which is one of the characteristics ofrapid solidification;(3)The dilution speed in the melt zone is low, which may lead to a good combination of the coating layer and the substrate metallurgically;(4)There is basically no restriction on the selection of coating materials, either metallic or non-metallic materials could be used. It is even possible to overlay a high-melting-point material on a low-melting-point metal to improve its specific properties.(5)The thickness of the coating layer could be selected in a large region. The coating area can also be accurately selected, which will lead to lower material consumption and, thereby, a lower capital cost.(6)The coating process can be automated, allowing special places that are usually inconvenient to handle to be processed.

In this paper, the manufacturing process of the nuclear-grade 316L stainless steel plate with a thickness of 40 mm was introduced for the first time. The plates manufactured with these procedures will be the base sample used for the following coating process. High-frequency laser welding with low laser power was then applied to melt a thin Fe-10Cr-4Al-RE layer on a plate made of austenitic 316L stainless steel. In order to enhance the welded samples’ mechanical properties, aiming to bear the nuclear-grade bending tests, the welding strategy and welding parameters were optimized based on metallurgical analyses and mechanical tests performed on the welded samples.

## 2. Material and procedures

### 2.1. Preparation of the Nuclear-Grade 316L Plate

The possibility of welding a protective Fe-10Cr-4Al-RE layer on a nuclear-grade SS316Lplain plate was studied. The plain plate with a thickness of 40 mm was manufactured according to the design of the reactor vessel wall of the China-Initiated Accelerator-Driven System (CiADS). The plain plate was hot rolled from the bulk nuclear-grade 316L steel manufactured under the standard of BTXG 018-2011 and qualified to the specifications listed in Table 1.

The Fe-10Cr-4Al-RE weld wire was imported from Kanthal AB, which was reported to be able to tolerate at least 1760 h corrosion from static liquid Pb at 900°C, thanks to the formation of a stable alumina protective layer at the contact surface when the oxygen level in liquid Pb was properly controlled [16]. The Fe-10Cr-4Al-RE alloy was imported as weld wires with 1.2 mm diameter.

### 2.2. FeCrAl Alloy Filler Material

The process of overlaying an FeCrAl thin layer on a plain plate made of nuclear-grade 316L stainless steel was described. The weld wire was provided by Kanthal AB, and is described in Table 2.

The elementary composition of the weld wire was analyzed using the Leco CS600 Carbon/Sulphur analyzer and ICP 725-ES inductively coupled plasma—optical emission spectroscopy following the ASTM-A751 standard procedures. The measured elementary vector is listed in Table 3, from which we can notice that the Al and Cr contents are relatively high, to enhance the formation of a protective oxide layer at the contact surface with the high-temperature Pb-based coolant.

### 2.3. Welding Processes

The overlay process was performed using both Ar-protected TIG welding and laser welding in individual processes, in order to determine the most economic, efficient and qualified coating strategy. A Panasonic YC400TX DC power source was used in the manual TIG welding process. Since laser welding requires more precise control of both the laser power and weld wire feeding speed, a laser welding station was proposed, consisting of a control cabinet, an MFSC 3000X single-module continuous-wave (CW) fiber laser generator, a WF007A weld wire feeding machine, a FANUC mechanical arm with a laser head, and a water cooling system, as shown in Figure 1.

### 2.4. Experimental Procedures and Results

#### 2.4.1. Manual TIG Welding and Results

Manual Ar-protected TIG welding was performed with the Fe-10Cr-4Al-RE weld wire in its original size, being equal to 1.2 mm in diameter. The thickness of the welding layer is around 3 mm, consisting of three sub-layers, as shown in Figure 2. The welding parameters are listed in Table 4. As shown in Figure 2, the welded surface is fairly rough and twisted, which was considered to be mainly caused by the significant melting and cooling shrinkage incurred by the high welding temperature.

Elementary compositions of the melt zone were analyzed with the OBLF QSG750 spark spectrometer with Ar gas, and are summarized in Table 5. The testing sample was polished using sandpaper with a mesh size of 80, and cleaned with high-purity alcohol. The pure Ar gas (purity of 99.999%) was supplied to the facility with apressure of 3 bar and a flowrate of 800 L/h.

Micro-hardness tests were performed for different layers of the welded sampleusing the Wilson Tukon 2100B micro-hardness tester, and by following the standard procedure GB/T-2654.The results are listed in Table 6. We observed a reduction in hardness from the welded layer to the base material. This phenomenon was considered to be mainly caused by the difference between the elementary compositions of FeCrAl weld wire and the austenitic stainless steel base material, as can be noticed from Table 1, Table 3 and Table 5. The hardness of the melt zone is higher than that of the base material, but lower than the FeCrAl weld wire, which was considered to be caused by the existence of Ti (formed during the hardening phase) and quenching from forced cooling.

After polishing the welded surface and performing the liquid penetrate test (PT), long cracks (as circled in Figure 3) penetrating the whole welding layer can be noticed in the central part of the test plate. This confirms the limitation of applying traditional manual TIG welding and, therefore, the necessity of developing an upgraded welding strategy.

In addition, side bending tests were performed on the first round of TIG welded samples. Significant cracks can be observed (Figure 4), which can mainly be explained by the huge columnar grains, which cross the whole welded layer, as shown in Figure 5. The side bending tests were performed with a 30-ton bend tester (GW40F) following the ANSI/AWS B4.0-2000 standard procedures. The welded plates were then cut into test specimens with a size of 150 × 40 × 10 (mm).

#### 2.4.2. Laser Welding Process and Results

Since our experience of welding an FeCrAl layer on a stainless steel plate is quite limited, information from open literature was collected as a reference plan. Firstly, parameter tests were conducted to obtain a set of welding parameters that can prevent the formation of cracks during welding processes, as described above. Secondly, bending tests were conducted to check its suitability for bending.

After a series of trial tests, a proper set of welding parameters that can prevent the formation of cracks after welding processes was proposed, as listed in Table 7. Using this set of parameters, a test plate coated with Fe-10Cr-4Al-RE was manufactured, as shown in Figure 6. Side bending tests and metallographic inspections were then performed on the test plate. The test results show quite limited improvement in the mechanical behaviors of the test samples using this set of welding parameters. There are still many cracks that can be observed, as shown in Figure 7. This is mainly due to the size and the shape of grains inside the overlay layer, as can be noticed in Figure 8.

It was reported in this author’s former articles that cracking during bending tests could be solved by reducing the grain size in welded samples, using a thinner weld wire with a 0.2 mm diameter and adjusting the welding parameters correspondingly. The laser power was set at a very low level. As discussed above, a surface layer thicker than 0.5 mm should be fully sufficient for corrosion resistance. Therefore, the thickness of the coating will not be a key problem to be considered during welding parameter optimization. The growth of crystal grains depends on the melting time at high temperature, so welding parameters can be optimized by reducing the heat input and increasing the welding speed to enhance surface cooling, and by matching the feeding speed of the weld wire to obtain a thinner coating layer with smaller grains inside.

After a series of tests, a set of optimized parameters were obtained: the laser power was set at 1500 W, the welding speed was set at 18 mm/s, the wire feeding speed was set at 100 cm/min, and the defocusing amount was set at +60 mm. The second test plate was overlaid using this set of parameters, as shown in Figure 9, and the side bending test and metallographic inspection were carried out against this sample plate, as shown in Figure 10 and Figure 11.

## 3. Discussion

Judging from the crack direction in the sample plate prepared with TIG welding, the source of cracks is close to the fusion line, and it will extend upward along the grain boundaries in the weld layer. Feather-like upper bainite was found in the dilution zone near these crack source points, as shown in Figure 5. The formation of cracks was then considered to be mainly caused by the brittleness of the micro-structure in the welding layer and the concentration of thermal stress. The production of upper bainite was considered to be mainly related to the chemical composition in the weld dilution zone, as well as the low interlayer temperature incurred by insufficient preheating. The welded line mainly consists of ferrite and granular brittle precipitates, which enables its plasticity and, thus, causes significant expansion.

The welding layer obtained from the laser welding using the first set of parameters is about 1.0 mm thick. The surface of this layer contained pores and fairly coarse grains, as can be observed in Figure 8. During the side bending tests, multiple cracks appeared and the crack depth exceeded more than half of the weld layer. After optimization of the laser welding parameters, the thickness of the coating layer was polished and tested. No defects could be found after PT tests, and the remaining coating layer was reported to be around 0.6–0.7 mm thick (as shown in Figure 10). As can be noticed from Figure 11, much finer columnar grains were reported to appear at the surface of the welding layer, and no obvious pores or feather-like upper bainite could be found, which could be used to explain its better mechanical properties when compared to the samples prepared using manual TIG welding. Moreover, as can be observed from Figure 12, no visible cracks were found after the side bending tests, which can preliminarily prove the feasibility of improving the mechanical performances of the sample plate by reducing the thickness of the welded layer and refining the grains inside it. This could be realized by properly adjusting the operational parameters of laser welding processes.

## 4. Conclusions

In order to obtain a proper welding strategy to overlay an FeCrAl alloy on a nuclear-grade austenitic 316L stainless steel plate, both traditional TIG welding and high-frequency laser welding were investigated and optimized, with the aim of surviving the nuclear-grade side bending tests, to match the conditions that could possibly be incurred by temperature changes or irradiation during the operation of CiADS. After the series of laser surfacing tests performed using Fe-10Cr-4Al-RE weld wire described in this article, the following conclusions can be preliminarily obtained:The flatness, straightness, and smoothness of the weld bead from the laser welding discussed in this article are far better than those of the argon arc welding stated above;Upgraded welding parameters can provide refined grains in the welded layer with a size of 50–300 μm;The sample plates with laser weld overlay were reported to be able to pass the nuclear-grade side bending test. This preliminarily proves the possibility of applying this set of welding parameters to overlay an Fe-10Cr-4Al-RE alloy on the inner surface of a lead-cooled fast reactor vessel;Operational parameters of the laser welding process will have significant impacts on the microstructures of a welding layer, especially the grain size within it, which will affect the mechanical properties of welded samples;The high-frequency low-power laser surfacing technology proposed in this article is suitable for producing a thin coating layer, comparable to that of the traditional TIG/MIG welding techniques. The thickness of a single welding layer can reach 1 mm and still be able to provide satisfying surface quality and mechanical properties.

The high-frequency low-energy laser welding system proposed in this article has been preliminarily proven to be suitable to overlay an Fe-10Cr-4Al-RE protective layer on a nuclear-grade austenitic 316L stainless steel plate, without causing cracks after welding or side bending, thanks to the refined grain structures in the melt zone. Since this kind of welded plate could be used to manufacture the wall of the CiADS main vessel, irradiation effects on the micro-structures of these plates should also be investigated by performing irradiation tests, and such tests are already planned by the authors.

## Figures and Tables

**Figure 1 materials-15-03541-f001:**
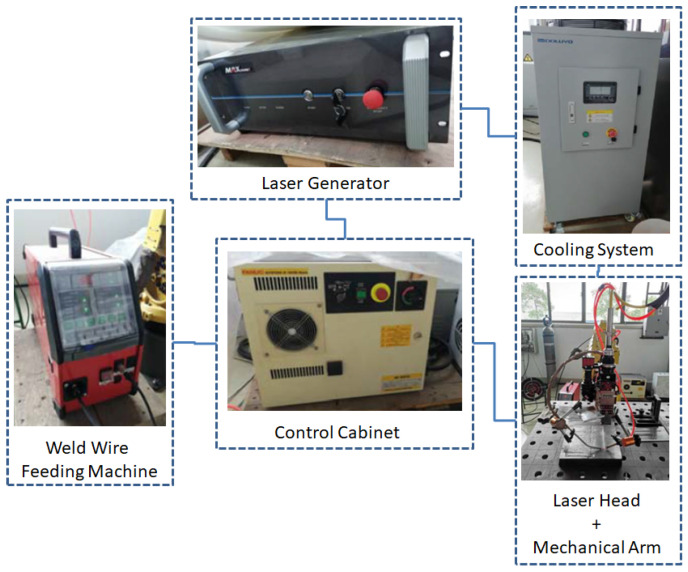
The layout of the laser welding station.

**Figure 2 materials-15-03541-f002:**
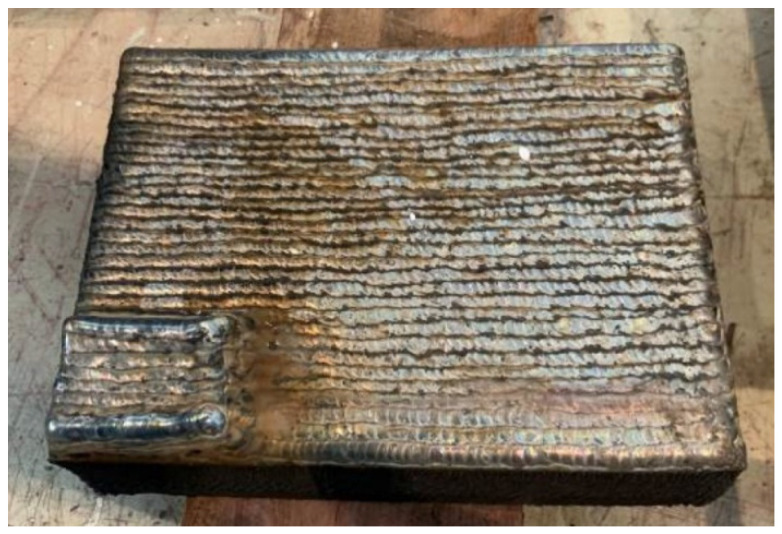
Plain plate welded by manual TIG welding technique.

**Figure 6 materials-15-03541-f006:**
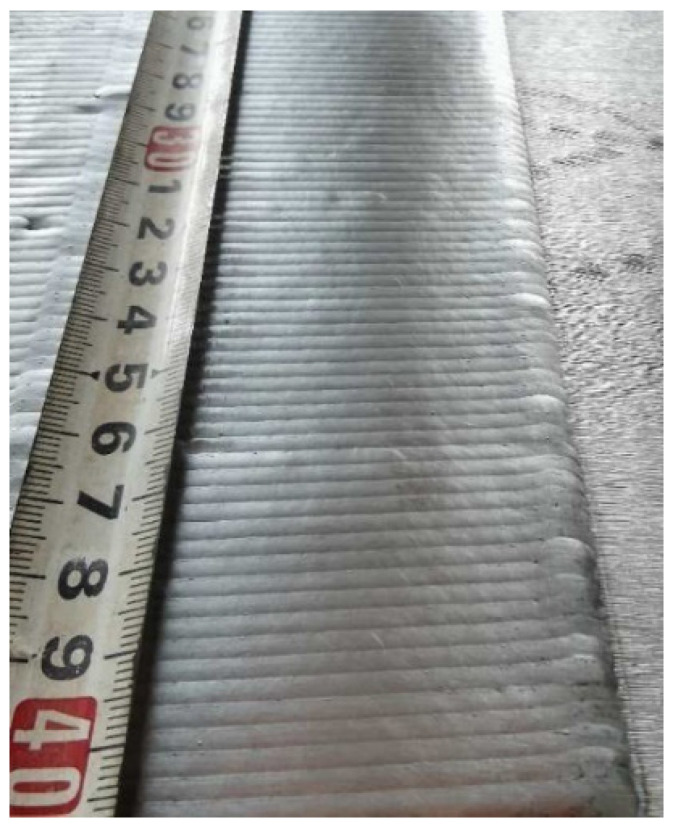
Appearance of first batch of laser weld overlay.

**Figure 7 materials-15-03541-f007:**
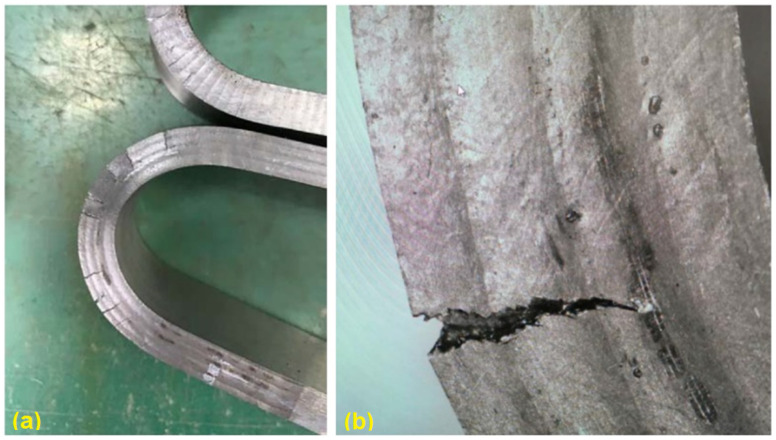
Cracks after side bending tests: test sample (**a**) and cracks (**b**).

**Figure 8 materials-15-03541-f008:**
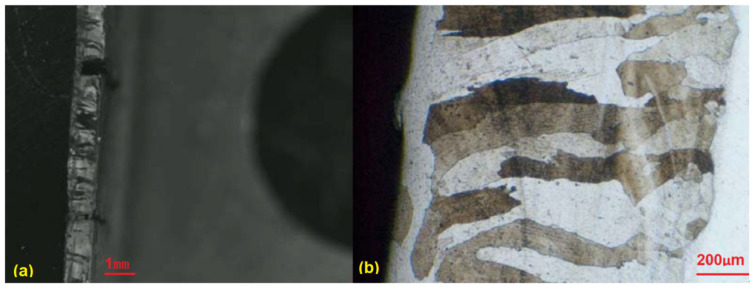
Metallographic analysis of the laser overlay plate (**a**) and the coating layer (**b**).

**Figure 9 materials-15-03541-f009:**
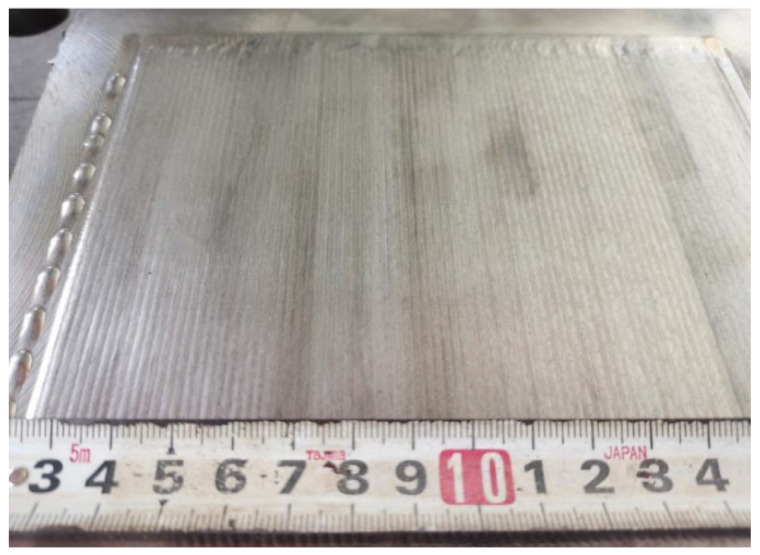
Appearance of the welded plate with upgraded parameters.

**Figure 10 materials-15-03541-f010:**
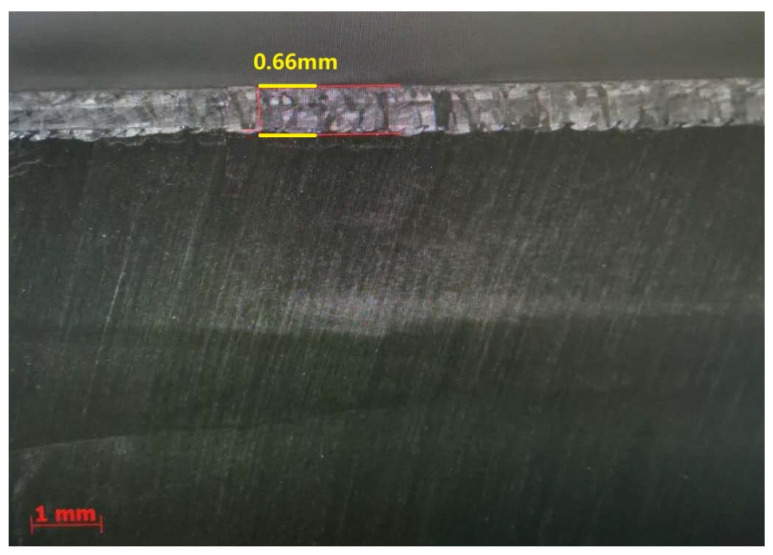
Macroscopic metallographic analysis of the welded plate with upgraded parameters.

**Figure 11 materials-15-03541-f011:**
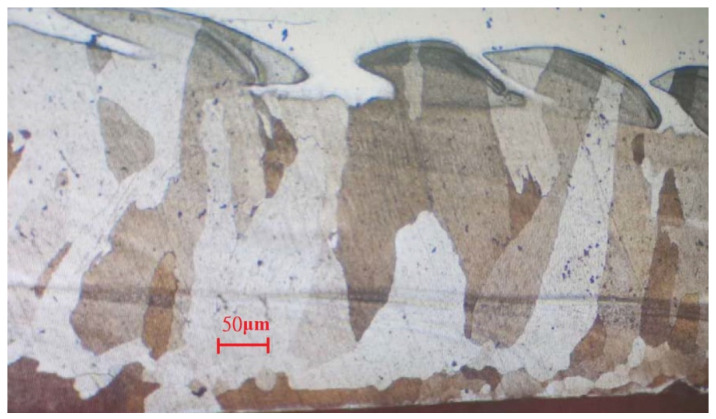
Microscopic metallographic analysis of the welded plate with upgraded parameters.

**Figure 12 materials-15-03541-f012:**
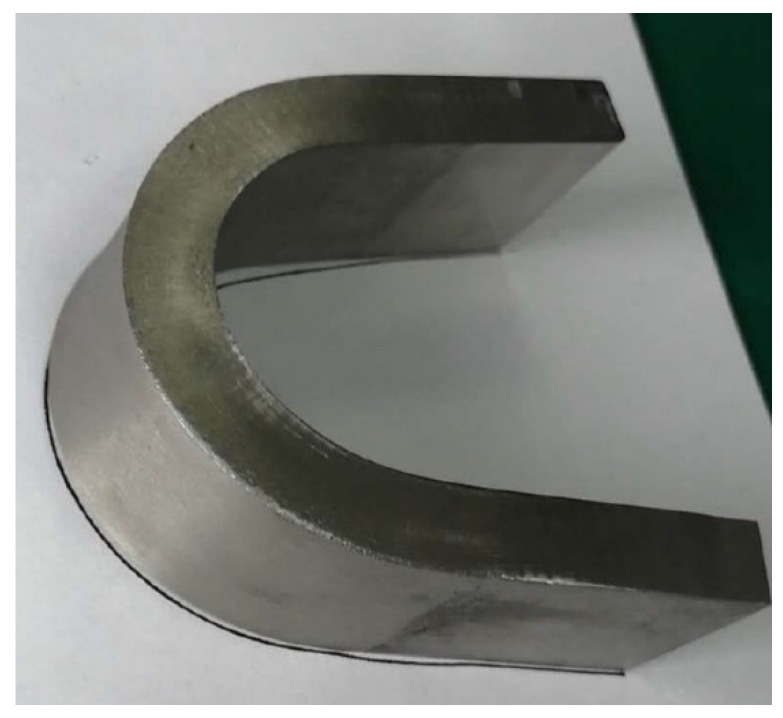
Result from the bending test with plunger size of 40 mm and bending angle of 180°.

**Figure 3 materials-15-03541-f003:**
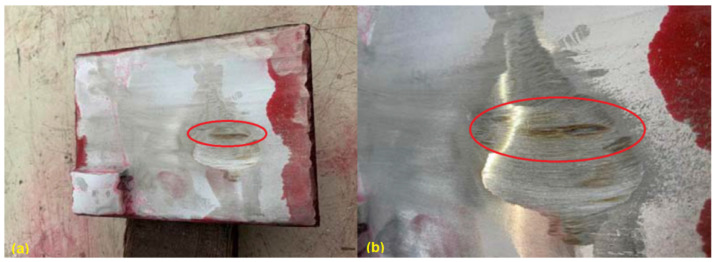
Appearance of the welded plate after liquid penetrate test (**a**) and cracks (**b**).

**Figure 4 materials-15-03541-f004:**
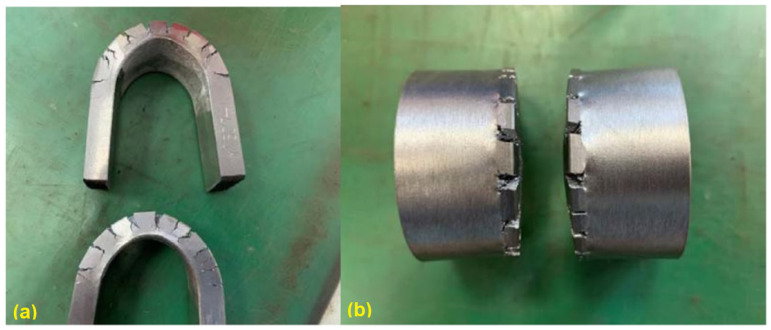
Manual TIG welded sample after side bending: side view (**a**) and top view (**b**).

**Figure 5 materials-15-03541-f005:**
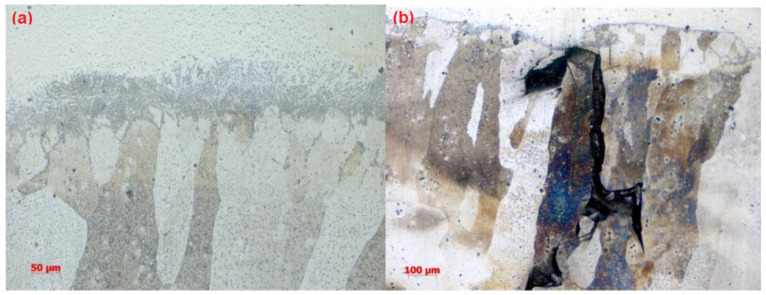
Metallographic analysis of the manual TIG welded sample: melting zone (**a**) and coating layer (**b**).

**Table 1 materials-15-03541-t001:** Specifications of the plain plate samples.

Chemical Composition													
C	Si	Mn		P	S		Cr	Ni		N	Co		Mo
22	43	144		17	2		1737	1227		600	4		233
**Mechanical Characteristics**													
Yield Stress, MPa			Tensile Stress, MPa			Elastic Length, %			Average Hardness, HBW			Grainess, ASTM grainess grade	
277			565			59			163.0			6.5–4.0	

**Table 2 materials-15-03541-t002:** Information of the weld wire.

Type	Diameter, mm	Batch No.	Manufacturer	Batch Weight, kg
Fe-10Cr-4Al-RE alloy	1.2	99324/142136	Kanthal AB/Sandvik	15

**Table 3 materials-15-03541-t003:** Elementary composition of weld wire (wt.%).

Ni	Cr	Al	C	Si	Mn
0.10	10.9	4.68	0.020	0.28	0.20
Fe	Ti	Cu	Nb + Ta	S	
Balanced	0.21	0.014	0.38	<0.001	

**Table 4 materials-15-03541-t004:** The first-round TIG welding parameters.

Current, A	Voltage, V	Welding Speed, mm/min	Heat Input, KJ/mm	Pre-Heating Temperature, °C
130	12	85–100	0.936–1.101	150
Max. interpass temperature, °C	Polarity	Tungsten bar size, mm	Protective gas	Ar gas flowrate, L/min
150–165	DCEN	3.0	99.999% Ar	13

**Table 5 materials-15-03541-t005:** Elementary composition of the melt zone (wt.%).

C	Si	Mn	P	S	Ni	Cr	Mo
0.018	0.30	0.22	0.012	0.003	0.11	9.37	0.029
V	Co	Cu	Ti	Nb	W	Al	/
0.026	0.013	0.022	0.21	0.46	0.15	3.37	/

**Table 6 materials-15-03541-t006:** Micro-hardness in different zones of weld overlay.

Parent metal area	185, 188, 181
Heat affected area	211, 209, 196
Hardfacing layer	265, 216, 200

**Table 7 materials-15-03541-t007:** Welding parameters to survive the welding processes.

Laser Power, W	Welding Speed, mm/s	Wire Feeding Speed, mm/min	Defocusing Amount, mm	Ar Flowrate, L/min
1600	12	130	+60	25

## Data Availability

Data sharing not applicable.

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
