# Peer review of "Influence of TIG and Laser Welding Processes of Fe-10Cr-4Al-RE Alloy Cracks Overlayed on 316L Steel Plate"

_materials, 2022, doi:10.3390/ma15103541_

Round 1
Reviewer 1 Report
The present form of manuscript is not suitable for consideration to publish in this journal. The presentation of this manuscript is look like a experimental report not as research article with novelty. Inappropriate data for reproducibility and insufficient information citation of recent articles to clarify the novelty of work is the main issues of this manuscript.
Author Response
Thanks for your comments on my current manuscript. This article summarized results from a series of experiments performed with different techniques and strategies, which surely revealed some explorations. We have added more citations in the introduction section and discussions in section 2 to prove the novelty of our work. We sincerely wish you to reconsider your decision on this manuscript. Thanks for your time and effort.
Reviewer 2 Report
Research article Titled "Study on the welding process to overlay Fe-10Cr-4Al-RE alloy on an austenitic stainless steel plate" is critically reviewed. Following modifications and observations are presented from the review:
- Introduction does not clearly reveal the current research in the filed of coating AL and Cr. Recent research related to added to introduction.
- Authors presented increase in hardness value after overlay using TIG process, justify why this so. In modern research author should present the micro hardness distribution along with surface hardness.
- The properties obtained after overlay, not justified in context to micro-structure finding.
- No detailed discussions of presented figures of results. In fig 12, scale bar is missing. use good quality figures to represent the results.
- Please add the discussion of your findings relating to physics.
- How you can conclude to grain size with optical micrograph, ."An upgraded welding parameters can provide refined grains in the welded layer with the size of 50-300μm;" Justify
Author Response
Introduction does not clearly reveal the current research in the field of coating Al and Cr. Recent research related to added to introduction.
- Thanks for your kind suggestions. I have already added relevant information into the "Introduction" section.
Authors presented increase in hardness value after overlay using TIG process, justify why this so. In modern research author should present the micro hardness distribution along with surface hardness.
- Thanks for your kind remind. I have added the micro hardness data at different layers of the welded samples. Please check.
The properties obtained after overlay, not justified in context to micro-structure finding.
- Thanks for your kind remind. Yes, you are right. We have tested the hardness of the samples prepared with manual TIG welding, from which we can observed the reduction of hardness from welded layer to the base material. This phenomenon was considered to be mainly caused by the difference between elementary compositions of FeCrAl weld wire and austenitic stainless steel base material. The hardness of melt zone is higher than that of the base material but lower than the FeCrAl weld wire, which was considered to be caused by the existence of Ti (forming hardening phase) and quenching from forced cooling.
No detailed discussions of presented figures of results. In fig 12, scale bar is missing. use good quality figures to represent the results.
- Thanks for your kind remind. I have added the scale bar and corresponding discussions into Section 2.4.2. Please check.
Please add the discussion of your findings relating to physics.
- Thanks for your kind suggestions. I have already added more detailed discussions in section 2.4.
How you can conclude to grain size with optical micrograph, ."An upgraded welding parameters can provide refined grains in the welded layer with the size of 50-300μm;" Justify
- The software accompanied with the micro metallographic analyzer we used in this study helped us to measure and summarize the distribution of grain size in cross section displayed in Fig. 11.
Reviewer 3 Report
This paper deals with the coating of FeCrAl material on 316L stainless steel plate with two welding processes: manual TIG welding and automated laser welding processes. The results show that the laser welding is the best process to overlay the filler metal on the 316L substrate by offering less surface crack after shaping the welded samples. The results and their interpretation sound good but the paper cannot be accepted in this form. Some details need to be improved for publication. Therefore, I recommend major revision for this article. To improve the quality of the paper, I have the following suggestions:
1) The title must be changed to: “Influence of TIG and Laser welding processes of Fe-10Cr-4Al-RE alloy cracks overlayed on 316L steel plate”
2) The abstract must be expended. It needs to summarize the objective of the study, the methodology used and the results found very directly and clearly.
Moreover, “we” and “our” must not be used in all the text.
3) In the introduction, some literature review is missing regarding to the different welding processes and their influences on the welding materials. Moreover, the end of the introduction must be rewritten by exposing what authors are going to do, better than the actual presented one (without exposing section 2.., section 3, …)
4) My main concern is the structure of the paper. I suggest to authors to have the following structure:
“ 2) Material and procedures
2.1) 316L plate
2.2) Filler metal alloy or 2.2) FeCrAl alloy filler material
2.3) Welding processes
2.4) Experimental procedure
3) Results and discussion”
In section 2.4, all the measured processes and the sample preparation-observation must be included
5) In current 3.1 “the major merits of Ar” could be in introduction (see comment 3)
6) Under table 4, how elementary compositions of the melt zone were analyzed? What was the process and the setting parameters?
7) What was the equipment used to perform the microhardness?
8) Current Fig. 3 is not relevant. It could be better to have a cross-section view of the bead to show the bead, the HAZ and the substrate. Moreover, in current Fig. 3 an arrow is missing to show the “long crack”. Same remark for Fig.8.
9) Why some bending tests have been performed? What do authors would like to show. It must be introduced better (see comment 4)). Moreover, what are the size of the bending samples? How are they bent (equipment, setting parameters…)?
10) For metallographic observation, samples mut be polished and etched. What are the polishing procedure and the etching procedure and solution (see comment 4))?
11) Current Fig. 10 is not relevant. It can be deleted. Quality of Fig 11 is poor: the distance in red cannot be seen correctly.
12) The conclusion must be rewritten and must be improved by recalling the context in the beginning It needs to expose relevant conclusions with respect to the results obtained and also the cross-results from discussion. The conclusion must be opened by announcing the prospects at the end.
Author Response
1) The title must be changed to: “Influence of TIG and Laser welding processes of Fe-10Cr-4Al-RE alloy cracks overlayed on 316L steel plate”
- Thanks for your kind suggestion. Yes, you are right, this title could be more suitable for this manuscript.
2) The abstract must be expended. It needs to summarize the objective of the study, the methodology used and the results found very directly and clearly.
- Thanks for your kind suggestion. I have added more relevant information into the abstract as you suggested.
Moreover, “we” and “our” must not be used in all the text.
- Thanks for this remind. I have rephased them.
3) In the introduction, some literature review is missing regarding to the different welding processes and their influences on the welding materials. Moreover, the end of the introduction must be rewritten by exposing what authors are going to do, better than the actual presented one (without exposing section 2.., section 3, …)
- Thanks for your kind suggestions. I have added supplementary paragraphs and rewritten the final paragraph in the "Introduction"
4) My main concern is the structure of the paper. I suggest to authors to have the following structure:
“ 2) Material and procedures
2.1) 316L plate
2.2) Filler metal alloy or 2.2) FeCrAl alloy filler material
2.3) Welding processes
2.4) Experimental procedure
3) Results and discussion”
In section 2.4, all the measured processes and the sample preparation-observation must be included
- Thanks for this thoughtful suggestion, which surely makes my manuscript better organized.
5) In current 3.1 “the major merits of Ar” could be in introduction (see comment 3)
- Thanks for this suggestion. I have moved and rewritten this part into the "Introduction" section
6) Under table 4, how elementary compositions of the melt zone were analyzed? What was the process and the setting parameters?
- I have added relevant information into the manuscript (section 3). We used the QSG750 spark-spectrometer with Ar gas to analyze the elementary composition of the melt zone.
7) What was the equipment used to perform the micro hardness?
- I have added the equipment's information (Wilson Tukon 2100B) into the reviewed manuscript.
8) Current Fig. 3 is not relevant. It could be better to have a cross-section view of the bead to show the bead, the HAZ and the substrate. Moreover, in current Fig. 3 an arrow is missing to show the “long crack”. Same remark for Fig.8.
- Thanks for your suggestion. In this part of our research, we mainly want to check if the cooling process after welding will cause cracks in the welded layer. It should be more valuable to observe the size and location of the crack. Thus, we used the surface photo instead of the cross-section photo. Besides, we have marked the location of crack in current photo as you suggested. Really appreciate for that.
9) Why some bending tests have been performed? What do authors would like to show. It must be introduced better (see comment 4)). Moreover, what are the size of the bending samples? How are they bent (equipment, setting parameters…)?
- Thanks for your suggestions about the nuclear-grade bending tests. Actually, we want to prove the welded stainless steel plate could survive the bending treatment during manufacturing and swelling incurred by temperature change or irradiation.
10) For metallographic observation, samples must be polished and etched. What are the polishing procedure and the etching procedure and solution (see comment 4))?
- I have added supplementary information into the manuscript as you suggested. Thanks for your kind remind.
11) Current Fig. 10 is not relevant. It can be deleted. Quality of Fig 11 is poor: the distance in red cannot be seen correctly.
- Thanks for your kind suggestion. I have upgraded the figures as you suggested.
12) The conclusion must be rewritten and must be improved by recalling the context in the beginning. It needs to expose relevant conclusions with respect to the results obtained and also the cross-results from discussion. The conclusion must be opened by announcing the prospects at the end.
- Thanks for your kind suggestions. I have rewritten my conclusions. Please check.
Round 2
Reviewer 1 Report
Authors have improved manuscript significantly as per suggested concerns. The present form of manuscript can be accepted for publication.
Author Response
Really appreciate for your kind approval of my manuscript. All your suggestions will be kept in mind and recalled when writing my next publication.
Reviewer 2 Report
Manuscript has been sufficiently improved to warrant publication in Materials.
Author Response

(The authors gave the same response as above.)

Reviewer 3 Report
Authors have made great efforts to improve the quality of the paper. Nevertheless, I have the following suggestion:
1) Fig. 3, 4, 5, 7 and 8 has 2 illustrations. Authors must present it in this form: “a) and b)” and refer it in the text. The captions must be change also.
2) At the beginning of sections 2.1 and 2.2 authors must avoid “In this article” or “In this section”.
3) This article must have a “Discussion” section to expose your findings relating to physics. This article presented like that will have more impact for the scientific community and the scientific soundness.
Author Response
1) Fig. 3, 4, 5, 7 and 8 has 2 illustrations. Authors must present it in this form: “a) and b)” and refer it in the text. The captions must be change also.
Thanks for your kind suggestion. I have updated these figures and citations in article.
2) At the beginning of sections 2.1 and 2.2 authors must avoid “In this article” or “In this section”.
Thanks for this suggestion. I have removed them as suggested.
3) This article must have a “Discussion” section to expose your findings relating to physics. This article presented like that will have more impact for the scientific community and the scientific soundness.
Thanks for this kind remind. I have added this section into the manuscript.
